# Genotypic Characterisation and Antimicrobial Resistance of Extended-Spectrum β-Lactamase-Producing *Escherichia coli* in Humans, Animals, and the Environment from Lusaka, Zambia: Public Health Implications and One Health Surveillance

**DOI:** 10.3390/antibiotics13100951

**Published:** 2024-10-10

**Authors:** Maisa Kasanga, Márió Gajdács, Walter Muleya, Odion O. Ikhimiukor, Steward Mudenda, Maika Kasanga, Joseph Chizimu, Doreen Mainza Shempela, Benjamin Bisesa Solochi, Mark John Mwikisa, Kaunda Yamba, Cheryl P. Andam, Raphael Chanda, Duncan Chanda, Geoffrey Kwenda

**Affiliations:** 1Department of Pathology and Microbiology, University Teaching Hospitals, Lusaka 15101, Zambia; benji.solochi@gmail.com (B.B.S.); nraphaelchanda@gmail.com (R.C.); 2Department of Oral Biology and Experimental Dental Research, Faculty of Dentistry, University of Szeged, 6720 Szeged, Hungary; gajdacs.mario@stoma.szote.u-szeged.hu; 3Department of Biomedical Sciences, School of Veterinary Medicine, University of Zambia, Lusaka 10101, Zambia; walter.muleya@unza.zm; 4Department of Biological Sciences, State University of New York, Albany, NY 12222, USA; oikhimiukor@albany.edu (O.O.I.); candam@albany.edu (C.P.A.); 5Department of Pharmacy, School of Health Sciences, University of Zambia, Lusaka 15101, Zambia; steward.mudenda@unza.zm; 6Department of Pharmacy, University Teaching Hospitals, Lusaka 15101, Zambia; maikakasanga@gmail.com; 7Zambia National Public Health Institute, Ministry of Health, Lusaka 10101, Zambia; chizimuyjoseph@yahoo.com (J.C.); kaundayamba@gmail.com (K.Y.); 8Department of Laboratory Services and Research, Churches Health Association of Zambia, Lusaka 10101, Zambia; doreen.shempela@chaz.org.zm; 9Department of Pathology, Lusaka Trust Hospital, Lusaka 10101, Zambia; mjmwwiiksa@yahoo.co.uk; 10Centre of Excellence for Adult Infectious Diseases, University Teaching Hospitals, Lusaka 15101, Zambia; duncanchanda@gmail.com; 11Department of Biomedical Sciences, School of Health Sciences, University of Zambia, Lusaka 15101, Zambia; jaffekwenda@gmail.com

**Keywords:** *Escherichia coli*, ESBL, antimicrobial resistance, whole-genome sequencing, Zambia

## Abstract

Background: Extended-spectrum β-lactamases (ESBL) in *Escherichia coli* are a serious concern due to their role in developing multidrug resistance (MDR) and difficult-to-treat infections. Objective: This study aimed to identify ESBL-carrying *E. coli* strains from both clinical and environmental sources in Lusaka District, Zambia. Methods: This cross-sectional study included 58 ESBL-producing *E. coli* strains from hospital inpatients, outpatients, and non-hospital environments. Antimicrobial susceptibility was assessed using the Kirby–Bauer disk diffusion method and the VITEK^®^ 2 Compact System, while genotypic analyses utilised the Illumina NextSeq 2000 sequencing platform. Results: Among the strains isolated strains, phylogroup B2 was the most common, with resistant MLST sequence types including ST131, ST167, ST156, and ST69. ESBL genes such as *bla*_TEM-1B_, *bla*_CTX-M,_
*bla*_OXA-1_, *bla*_NDM-5_, and *bla*_CMY_ were identified, with ST131 and ST410 being the most common. ST131 exhibited a high prevalence of *bla*_CTX-M-15_ and resistance to fluoroquinolones. Clinical and environmental isolates carried *bla*_NDM-5_ (3.4%), with clinical isolates showing a higher risk of carbapenemase resistance genes and the frequent occurrence of *bla*_CTX-M_ and *bla*_TEM_ variants, especially *bla*_CTX-M-15_ in ST131. Conclusions: This study underscores the public health risks of *bla*_CTX-M-15_- and *bla*_NDM-5_-carrying *E. coli*. The strengthening antimicrobial stewardship programmes and the continuous surveillance of AMR in clinical and environmental settings are recommended to mitigate the spread of resistant pathogens.

## 1. Introduction

Antimicrobial resistance (AMR) is a global concern, with the World Health Organisation (WHO) reporting that the rise of antibiotic-resistant bacteria could render essential drugs ineffective for the treatment of common infections [1,2]. The misuse of antimicrobials in livestock feed [3], human medicine [4], and agriculture has significantly contributed to about 50–80% of the emergence and spread of AMR [5,6]. The WHO’s Global Report on AMR highlights alarming levels of bacterial resistance to commonly used antimicrobials worldwide, leading to ineffective treatment for common infections, such as urinary tract infections, respiratory tract infections, gastrointestinal infections, and bloodstream infections [5,7]. In the absence of successful preventative measures, it is projected that by 2050, AMR may become a leading cause of mortality globally. Estimates suggest that mortality directly attributable to AMR rose to 1.2 million in 2019, with projections indicating up to 10 million deaths annually by 2050 if appropriate measures are not implemented [5].

AMR poses a particular threat in sub-Saharan Africa (SSA), where the projected mortality rate is the highest at 23.5 deaths per 100,000 people [8]. In central SSA, the mortality rate attributed to AMR is 20.7 (14.9–27.7) per 100,000 individuals [9]. Countries like Zambia face a considerable public health challenge from AMR, though data on its full impact is limited. A 2019 study reported 3700 deaths directly attributable to AMR and 15,600 associated with AMR, ranking the country 191st globally [8,10,11].

Multidrug-resistant (MDR) infections, especially from pathogens such as *Escherichia coli*, pose significant risks due to limited treatment options. These infections lead to severe conditions and therapeutic challenges, particularly in paediatric patients and intensive care unit (ICU) patients [12]. The prevalence of MDR pathogens has risen, with increasing resistance rates observed in recent years [13]. The WHO recognises MDR bacteria as a global health threat, stressing the need for timely detection and treatment to reduce high mortality and morbidity rates [14].

*E. coli* is a member of the ESKAPE group of pathogens, including *Enterococcus faecium*, *Staphylococcus aureus*, *Klebsiella pneumoniae*, *Acinetobacter baumannii*, *Pseudomonas aeruginosa*, and *Enterobacter* spp.—and is ranked among the top six most deadly MDR microorganisms [15]. It is usually associated with the production of extended-spectrum β-lactamases (ESBL), and its isolates commonly harbour plasmids containing ESBL genes, which confer resistance to various antibiotic classes, including tetracyclines, β-lactams, quinolones, macrolides, sulfonamides and aminoglycosides [4].

ESBL production has become a global problem in recent decades, primarily associated with *E. coli*, *Klebsiella* species, and other members of the *Enterobacteriaceae* family, with recent studies reporting a rise in β-lactamase production in these organisms [15]. In SSA, ESBL-producing *E. coli* is increasingly recognised as a significant pathogen in invasive infections, such as bacteremia, posing a severe threat to clinical practice. ESBLs are plasmid-borne enzymes that hydrolyse β-lactam rings in penicillins, broad-spectrum cephalosporins, and monobactams, leading to the loss of bactericidal activity [16]. A Zambian study previously reported a high positivity rate (31.8%) for ESBLs in *E. coli* isolates from clinical and environmental specimens [16]. Another study from Indonesia found an 85% prevalence of ESBL-producing *E. coli* [17,18]. However, little is known about the horizontal gene transfer of resistant bacteria in this region [9,10,14,19]. In Zambia, there is growing concern over rising AMR rates, particularly regarding frequently used empirical treatments [20].

The misuse of antibiotics creates unnatural selective pressure in clinical and natural environments, harming human and animal health [4,21]. Bacterial resistance arises naturally from genetic mutations or the acquisition of new resistance genes through horizontal gene transfer mechanisms [22]. Resistance to third-generation cephalosporins in bacterial pathogens is linked to ESBL gene acquisition, with ESBL-producing enteric bacteria colonising 14% of the global population and increasing by 5.4% annually [23,24].

The global spread of *bla*_CTX-M-15_, the predominant CTX-M variant, is associated with other variants such as *bla*_CTX-M-14, -15, -27,_ and *bla*_CTX-M-55,_ indicating the potential for transmission between humans and animals [25,26]. A study in the Netherlands reported that *bla*_CTX-M 15_ was the most prevalent (31.3%) ESBL type, with similar findings in Sudan, Egypt, and Saudi Arabia [27]. Recent studies in Zambia have also identified *bla*_CTX-M_ genes, particularly *bla*_CTX-M-15,_ in *E. coli* isolates from urinary tract and bloodstream infections [26]. Further studies revealed similarities in plasmid backbones among ESBL-producing *E. coli* strains from different lineages, including strains isolated from animals such as chickens, cattle, and swine, as well as humans [28]. The multiple transmission levels of strains, plasmids, and possibly genes within Zambia require the investigation of the key drivers using a “One Health” approach to combat the current ESBL epidemic.

Therefore, this study sought to analyse the genetic profiles of ESBL-producing *E. coli* strains in Lusaka, Zambia, to assess their distribution, transmission dynamics, and potential public health implications. We hypothesised that the genetic characteristics of *E. coli* isolates from clinical and environmental sources are similar, underscoring the urgent need for coordinated efforts to address this growing threat.

## 2. Results

### 2.1. Prevalence of Antimicrobial Resistance

A total of 980 samples were analysed, comprising 480 clinical and 500 environmental samples. Overall, 632 samples (332 clinical and 300 environmental samples) (64.5%) tested positive for *E. coli*. Of these, 31.8% (201/632) of *E. coli* isolates (141 clinical and 60 environmental) were identified as ESBL-producers on the VITEK compact system. Out of these isolates, 34.8% (49/141) and 15% (9/60) from clinical sources (urine, pus, and blood) and environmental sources (water, meat, medical equipment, and vegetables), respectively, were subjected to Whole-GenomeSequencing (WGS).

The *E. coli* isolates were susceptible to amikacin (AMK) (100%), ertapenem (ERT) (94.8%), imipenem (IMP) (96.6%), meropenem (MEM) (96.6%), ciprofloxacin (CIP) (91.4%), levofloxacin (LEV) (86.3%), amoxicillin-clavulanate (AMC) (69%), cefazolin (KZ) (91.4%), ceftriaxone (CRO) (91.4%), and gentamicin (CN) (68.9%). All ESBL isolates exhibited extremely high resistance to ampicillin (94.8%), followed by nitrofurantoin (NIT) (48.3%), trimethoprim-sulfamethoxazole (SXT) (41.3%), and tetracycline (TET) (41.3%) (Figure 1).

Ceftriaxone (CRO), ampicillin (AMP), amoxicillin-clavulanate (AMC), cefuroxime (CXM), ciprofloxacin (CIP) and trimethoprim-sulfamethoxazole (SXT) were linked to multiple phylogroups (A, B2, F, C, D G), with phylogroup B2 being the most common for both environmental and clinical isolates (Table 1).

The distribution of phenotypic antibiotic resistance among clinical and environmental *E. coli* isolates in Table 2; no significant differences was observed among the gene distribution in clinical and environmental *E. coli.*

The distribution of AMR genes and β-lactamase production in ESBL-producing *E. coli* strains from clinical and environmental isolates (Figure 2). Resistance patterns between CRO-AMP-CXM-CIP and CRM-AMP-AMC-CXM and CIP SXT showed a strong correlation with *bla*_CTX-M 15_, while susceptibility was observed for *bla*_OXA-10_, *bla*_NDM_5_, *bla*_CTX-M 14_, *bla*_TEM 141_, *bla*_TEM-206_, *bla*_TEM-216_ and *bla*_TEM-214_.

Quinolone (10%) AMR genes were the most prevalent group, followed by resistance genes for β-lactams, aminoglycosides, genes encoding for efflux pumps (approximately 7.5% respectively), resistance genes for fosfomycin, sulphonamide, tetracycline, trimethoprim, macrolides (approximately 5.0% respectively), colistin, mercury, bleomycin, phenicol, tellurium, silver, nitrofuran, rifamycin, and arsenic (approximately ≤ 2.5%, respectively) (Figure 3).

The distribution of phylogenetic analysis of E. coli isolates across clinical and environmental demonstrated a diverse distribution pattern among different phylogroups. The findings indicate that phylogroup B2 (59.2% vs 66.7%) was the most prevalent, followed by phylogroup A (20.4% vs 11.1%), B1 (8.2% vs 11.1%) and C 2.0% vs 11.1%) respectively. The other phylogroups such as D (4.1%), D, B2 (2.0%), F (2.0%) and G (2.0%) were common in clinical isolates and exhibited lower frequencies respectively (Figure 4).

### 2.2. Distribution of MLST Sequence Type and O-Antigen Serotype

The O-antigen serotype and sequence type (ST) were further identified for bacterial typing to differentiate strains. A total of 14 distinct STs were identified, ST131 (n = 22, 37.9%), ST167 (n = 8, 13.8%), and ST69 (n = 4, 6.9%) being the most prevalent in clinical isolates, whilst ST131 (n = 4, 6.9%) and novel STs (n = 2, 3.5%) dominated environmental isolates (Figure 5). Overall, 11 different O-antigen serotypes were identified in Figure 5, with O25 (n = 30, 51.7%) and O101 (n = 7, 12.1%) being the most common in clinical isolates, and O25 (n = 5, 8.6%) and O101 (n = 2, 3.5%) being found in environmental isolates. The most prevalent strain across clinical and environmental samples was *E. coli* O25-ST131 (n = 21, 36.2%), followed by ST167-O101 and ST69-O25, each with 4 isolates (8.6%). ST167-O25 was identified in 3 clinical isolates (5.2%). The rest of the serotypes and sequence types exhibited low prevalence (1.7%) in clinical and environmental isolates.

#### 2.2.1. Prevalence of Resistance Genes for Trimethoprim, Sulphonamides, Tetracyclines, and Acquired Quinolone Resistance

The prevalence of trimethoprim resistance genes in *E. coli* was observed as follows: *dfrA*12 (13.8%), *dfrA*14 (17.2%), and *dfrA*17 (50.0%). There was a higher presence in clinical isolates (87.5%). The most prevalent resistance genes were *sul2* (72.4%) and *sul1* (63.7%), predominantly found in clinical samples (over 80%). *tet* (A) was the most prevalent tetracycline resistance gene (63.75%), with a high presence in clinical isolates (83.7%), while *tet* (B) was found in 94.1% of the clinical isolates. The acquired quinolone gene *aac (6′)-ib-cr5* was present in 32.7% of the isolates, and in 15.7% of environmental isolates, while *qnrS1* was found in 66.7% of environmental and clinical isolates. *qnrB1* (5.1%) and *qnrS1* (5.1%) were found exclusively in clinical isolates. The high prevalence of resistance genes in *E. coli* isolates, particularly those associated with macrolide-lincosamide-streptogramin B (MLSB) resistance, was notable. The *mdf(A)* gene was present in 100% of isolates, while *mph(A)* was found in 65.8%. Phenicol resistance genes *catA1* and *catB3* were detected in 15.4% of the isolates, and *floR* and *cmlA1* were found in 1.9% of the *E. coli* isolates (Table 3).

#### 2.2.2. Prevalence of β-Lactam AMR Genes in in ESBL-Producing *E. coli* Strains

In all ESBL-producing *E. coli* isolates from both clinical and environmental samples, only the *bla*_CTX-M-15_ gene was observed across a wide range of sources. It was present in 34.5% of isolates and 10.4% of pus, while in blood, water, meat, equipment, and vegetables its presence ranged from 1.7% to 5.2%. The *bla*_EC_ gene was found in 60.4% of urine isolates, 20.7% of pus isolates, and3.5% to 5.2% of other sources. The *bla*_OXA-1_ gene was prevalent in 25.9% of urine isolates. Additionally, two isolates (1.7%) were positive for the *bla*_NDM-5_ gene. The *bla*_CTX-M-14_, *bla*_CTX-M-55_ and *bla*_TEM_ genes were detected in 1.7%, 1.7% and 3.5% of the isolates (Table 4), respectively.

## 3. Discussion

The present study focused on the genetic diversity and AMR of *E. coli* in clinical and environmental isolates in Lusaka, Zambia. As a part of this investigation, the genomes of 58 ESBL-producing *E. coli* isolates were sequenced using WGS. The findings revealed that *E. coli* in this region has a growing pan-genome, indicating significant genetic diversity. The phylogenetic clustering, STs, and phylogroups of the *E. coli* isolates also displayed considerable alignment with global variation in resistance genes.

In Zambia, the misuse of antibiotics, particularly third-generation cephalosporins, has been reported to be a driver for the high resistance observed in humans, animals, and the environment [29,30]. The rise of antibiotic resistance in *E. coli* is a major public health concern, with studies indicating that resistance rates vary among antibiotics. Notably, this study found that about two-thirds of the *E. coli* isolates exhibited resistance to at least one antibiotic from multiple classes, with the highest rates observed for ampicillin, cefazolin, ciprofloxacin, tetracyclines, and sulfonamide/trimethoprim. In contrast, carbapenems showed low resistance rates, and no resistance to amikacin was detected in any of the isolates. These findings are consistent with a study conducted in Yemen, which reported resistance rates greater than 90% for ampicillin and cefazolin in clinical isolates [31]. Similarly, a study in Saudi Arabia reported high resistance to nitrofurantoin (85.4%) and ampicillin (over > 80%) in urine isolates [32]. A study in Turkey also reported high resistance rates to ampicillin (87.3%) and cefuroxime (71.6%) in patients with urinary tract infections (UTI), though lower rates were observed in outpatients [33]. The MDR pattern observed in this study could be attributed to factors commonly associated with low and middle-income countries. These factors include the misuse of antibiotics, the absence of effective antibiotic stewardship programmes, limited access to quality healthcare, and the inadequate regulation of antibiotics [9]. This finding aligns with several other studies, including those conducted in Nigeria, which reported the occurrence of MDR and extensively drug-resistant (XDR) *E. coli* in poultry [34]. A study in Benin reported resistance rates to ciprofloxacin (91.3%), levofloxacin (86.3%), and sulfonamide (58.6%) [35]. These resistance rates call for implementation of strategies to address AMR in a One Health approach.

This finding that antimicrobials are frequently employed in Zambia poultry farming is consistent with reports of their widespread availability, especially in the absence of robust antimicrobial stewardship programmes [36]. A study by Fenollar-Penadés et al. (2024) [37] showed a lower resistance rate to ampicillin (76.9%) in faecal isolates from breeding hens, which highlights the varying levels of AMR in different environments. The persistence of resistant *E. coli* on farms also raises concerns about cross-contamination within and between flocks.

Globally, the prevalence of ESBL-producing bacteria has risen, with reported cases ranging from 33% to 91% [38]. In Southern Africa, ESBL genes found in isolates from human, animal, environmental, and hospital settings are increasingly common [39]. The most prevalent ESBL gene in this study was *bla*_CTX-M_, followed by *bla*_TEM_. A Tanzanian reported similar findings, with *bla*_CTX-M_ and *bla*_TEM_ observed in 88.1% and 51.1% of the UTI cases, respectively [40,41]. This trend has been supported by numerous studies globally, which have shown the predominance of the *bla*_CTX-M_ gene, ranging from 80% to 100% [22,39]. Studies in Bangladesh [42] and China [43] reported the prevalence of the *bla*_CTX-M_ gene to be 85.71% and 39.5%, respectively, while studies in Zambia have also demonstrated a high prevalence of *bla*_CTX-M_ in humans and animals [25,26]. *E. coli* isolates are positive for ESBL-encoding genes, *bla*_CTX-M_ and *bla*_TEM,_ suggesting that human, animal, and environmental isolates are distributing ESBL-producing *E. coli*.

This study identified the genes responsible for observed resistance phenotypes in *E. coli* in Zambia, both in clinical and environmental isolates. A study in Egypt found that the most prevalent ESBL genes were *bla*_TEM_ (64%), followed by *bla*_SHV_ (30%) and *bla*_CTX-M_ (22%) [44]. A study conducted among intensive care unit patients in Yemen found 100% of *bla*_TEM_ genes and 33.3% of *bla*_CTX-M_ genes [45]. Similarly, a study in Lebanon revealed the presence of *bla*_CTX-M_ (92%) and *bla*_TEM_ (86%) in isolates from wound infections [46], consistent with the findings of this study. The high prevalence of the *bla*_TEM_ and *bla*_CTX-M_ genes reported in our study and other study indicate the potential of bacteria to hydrolyse beta-lactam antibiotics. Hence, this requires urgent attention regarding strategies to promote rational use of antibiotics to reduce AMR. 

The widespread prevalence of *bla*_CTX-M_ among ESBL-producing bacteria can be explained by the high transfer efficiency of conjugative plasmids-carrying *bla*_CTX-M_ allele. This allele has been frequently reported as being most successfully transferred through horizontal gene transfer [47]. The rapid spread of the *bla*_CTX-M_ gene contributes to the unpredictable changes in the epidemiology of AMR, making it a public health concern [48]. Commensal *E. coli* may serve as a reservoir for these resistance genes, potentially facilitating human transmission. The high frequency of this gene in clinical and environmental isolates highlights the increasing threat that antibiotic-resistant bacteria pose to public health. Its ability to spread quickly contributes to rising rates of antibiotic resistance, complicating treatment options and posing significant challenges for infection control.

The continued relevance of *E. coli* as the primary pathogen linked to UTIs is connected to alterations in the prevalence of certain phylogroups, especially B2. Notably, bacterial isolates classified as part of the B2 phylogroup are commonly linked to infections outside the intestines, such as UTIs. The rise in frequency might correlate with a greater number of virulence-associated genes compared to the other phylogroups. This discovery is why almost half of the ESBL *E. coli* isolates in the present study belong to the phylogenetic group B2. All isolates were positive for O25b-ST131, an ST linked to a pandemic clone belonging to the B2 phylogroup. A study conducted in a Mexican hospital reported that the phylogroup B2 caused 54.4% of UTIs. Among these infections, 46.5% were found to be of the ST131-O25 clone [49].

The pandemic-distributed ST131 clone of the O25:H4 serotype significantly contributes to hospital- and community-acquired UTIs worldwide. In the present study, multiple extended-spectrum cephalosporin and fluoroquinolone resistance genes were identified in the *E. coli* ST131 strain, with further evidence suggesting that ST131-type *E. coli* may acquire resistance genes more effectively, thereby impacting treatment. The ESBL gene *bla*_CTX-M-15_ and ST131 were common in clinical and environmental isolates. The virulent ST131 *E. coli* is a predominant lineage among extraintestinal pathogenic *E. coli* (ExPEC) and has played a significant role in the global dissemination of *bla*_CTX-M-15_. It has been reported in human and non-human sources in Nigeria [22], Tunisia [50], and China [51].

Two novel MLSTs, ST13383 and ST10955, were identified in this study, harbouring several AMR genes such as aminoglycosides, *bla*_CTX-M 15_, *bla*_EC_ and *bla*_TEM-1_, sulphonamides, quinolones (*qnrS1*), and tetracyclines (*tet*[A] and *tet*[B]). Ciprofloxacin, a widely prescribed antibiotic in Zambia, may be linked to the carriage of *aac (6′)Ib-cr* and *bla*_CTX-M_ ESBL due to its high prescribing rate. Previous studies have reported the connection between quinolone resistance and sub-lineages of ST131. In a study by Castillo et al. (2024), [37] it was reported that *qnr*B (85.4%) and *qnr*S (24.4%) were higher in faeces from breeding hen chickens compared to the present study, which observed a lower prevalence. This difference may be attributed to differences in antibiotic usage practices or environmental conditions in breeding hen farms.

Over the past decade, the global dominance of short-term carbapenem resistance has fluctuated, with the prevailing ST shifting from ST131 (43.1%), which exhibits higher virulence but lower antibiotic resistance, to ST410 and ST167, which demonstrate higher resistance and slightly reduced virulence [52]. A study in Spain reported that ST131 strains accounted for 42.3%, followed by China (19.05%), the US (18.85%), and the UK (17.45%) [51]. These findings are consistent with the results in the current study, which showed a high prevalence of ST131 in both environmental and clinical isolates [53]. These results suggest that *E. coli* isolates could pose a risk to humans, animals, and the environment due to MDR.

This study found that phylogroup B2 was dominant in all isolates, followed by phylogroup A. However, in Mexico (35%), a report showed that the prevalence of phylogroup A isolated from patients with UTIs was higher than that of phylogroup D [54].

The present study demonstrated considerable clonal diversity among ESBL-producing *E. coli* strains. The emergence of multiple separate clonal groupings underscores the complex structure of their genetic relationships. This diversity facilitated the prevention of outbreaks due to the simultaneous presence of numerous populations. In line with the initial hypothesis, genetic characteristics were found to be similar in both clinical and environmental sources.

## 4. Materials and Methods

### 4.1. Study Site and Sampling

This study employed a three-tier cross-sectional design and was conducted between July 2022 and March 2023, involving humans, animals, and the environment. The human component was conducted at the University Teaching Hospital (UTH), a referral tertiary and highly specialised facility comprising five hospitals: the Adult Hospital, Children’s Hospital, Mother and Newborn Hospital, Eye Hospital, and Cancer Disease Hospital (CDH). UTH, with a capacity of approximately 1652 beds, serves as a national referral hospital for all ten provinces of Zambia. Lusaka Province, where UTH is located, has a population of approximately 3,079,964 [55] and an estimated household of 687,923 [56]. The animal and environment components were conducted within the UTH hospital environment and in sixteen sub-administrative districts (communities) of Lusaka Province.

A convenient sampling method was used to collect 980 clinical samples (urine, blood, pus, cerebrospinal fluid [CSF]) and environmental samples (borehole water, hospital equipment, fish, meat, and vegetables). A total of 58 isolates were selected from ESBL-positive isolates, which showed suspected MDR and non-ESBL isolates to be resistant to more than three classes of antibiotics for sequencing. Clinical samples were obtained from the microbiology laboratory, while environment samples were collected from the UTH environment and transported to the laboratory within three hours.

### 4.2. E. coli Isolation and Identification

The isolation of *E. coli* colonies from clinical (human) specimens was performed on Xylose Lysine Deoxycholate (XLD) agar (Oxoid Ltd., Basingstoke, Hampshire, UK), MacConkey agar (Oxoid Ltd., Basingstoke, Hampshire, UK), and Hichrome chromogenic UTI agar (HiMedia Laboratories Pvt. Ltd., Mumbai, India). Urine samples were directly inoculated onto Hichrome chromogenic UTI agar (HiMedia Laboratories Pvt. Ltd., Mumbai, India). All the cultured plates were incubated at 37 °C for 18 to 24h. The presumptive identification of the isolates was performed by plating the isolates on Eosin Methylene Blue (EMB) agar Oxoid Ltd., Basingstoke, Hampshire, UK) incubated further for 18 to 24 h at 37 °C.

The environmental samples were inoculated directly into buffered peptone water (BPW) (Oxoid Ltd., Basingstoke, Hampshire, UK) for enrichment at 37 °C for 3 h. A loopful of the culture was then transferred onto CHROMagar™ ECC (HiMedia Laboratories Pvt. Ltd., Mumbai, India) agar plates at 37 °C for 18 to 24 h to isolate *E. coli*. A series of biochemical tests, including triple sugar iron (TSI) agar, lysine iron agar (LIA), Simmons citrate agar (SCA), and sulphide indole motility (SIM) agar (Oxoid, Basingstoke, UK), were used for the presumptive phenotypic identification of the isolates. Biochemical confirmation of the *E. coli* isolates was performed using the Vitek II System (bioMérieux SA, Marcy-l’Étoile, France).

### 4.3. Antimicrobial Susceptibility Testing

The *E. coli* isolates studied were subjected to antimicrobial susceptibility testing according to the Kirby–Bauer disk diffusion method specified by the Clinical and Laboratory Standards Institute (CLSI) [57,58]. The antibiotics (Oxoid, Basingstoke, UK) tested included nalidixic acid (NAL) (30 μg), amoxicillin-clavulanate (AMC) (60 μg), amikacin (AMK) (30 μg), ceftazidime (CAZ) (30 μg), ampicillin (AMP) (25 μg), ceftriaxone (CRO) (30 μg), chloramphenicol (CHL) (30 μg), cefepime (FEP) (30 μg), ciprofloxacin (CIP) (10 μg), levofloxacin (LEV) (10 μg), meropenem (MEM) (10 μg), ertapenem (ETP) (10 μg), imipenem (IMP) (10 μg), nitrofurantoin (NIT) (300 μg), and trimethoprim-sulfamethoxazole (SXT) (5 μg). All the plates were incubated at 37 °C for 24 h. Susceptibility testing of the isolates was also performed on the Vitek II System (bioMérieux SA, Marcy-l’Étoile, France). The choice of antimicrobial drugs was based on the common drugs used for the empirical treatment of *E. coli* infection in Zambia. Results were interpreted as susceptible, intermediate or resistant based on the Clinical Laboratory Standards Institute (CLSI) guidelines [57]. *E. coli* isolates that exhibited resistance to at least three classes of antimicrobial agents were classified as multidrug-resistant (MDR) [59]. *E. coli* ATCC 25922 was used as a reference strain.

ESBL detection was performed by using the double-disk synergy test between clavulanic acid and extended-spectrum cephalosporins (ceftazidime and cefotaxime) on Müeller–Hinton agar plates [60] after incubation at 37 °C for 24 h. ESBL-positive isolates on the Vitek II System (bioMérieux SA, Marcy-l’Étoile, France) were automatically flagged by the system.

### 4.4. Whole-Genome Sequencing and Bioinformatics Analysis

Genomic DNA was extracted using the QIAamp DNA Mini Kit (QIAGEN, Hilden, Germany) and quantified using the Qubit Fluorometer (ThermoFisher Scientific, Waltham, MA, USA) following the manufacturers’ protocols.

Libraries were prepared using the Nextera DNA Flex Library Preparation Kit (Illumina Inc., San Diego, CA, USA) following the manufacturer’s instructions. They were then sequenced on an Illumina NextSeq 2000 sequencer using a 2 × 150 paired-end protocol (Illumina Inc., San Diego, CA, USA) at the National Institute for Communicable Diseases in Johannesburg, South Africa. The quality of the raw sequence reads was assessed using FastQC v.0.11.9 [61]. Fastp was used to remove adaptors and low-quality reads.

The filtered reads were used as input for de novo genome assembly using the Shovill v.1.1.0 package (https://github.com/tseemann/shovill; accessed on 22 November 2023) and incorporated the SPAdes v3.14.1 algorithm [62]. The quality of genomes was assessed using QUAST v.5.0.2 [63]. The threshold for high-quality genomes was set at <200 contigs and an N50 > 40,000 bp. Furthermore, CheckM v.1.1.3 [64] was used to determine the level of completeness and contamination of the assemblies. A threshold of greater than >90% was used for completeness, while a threshold of less than 5% was used for contamination of less than 5% [63]. Bactinspector v0.1.3 (https://gitlab.com/antunderwood/bactinspector; accessed 11 December 2023) was used to identify the sequenced isolates as *E. coli*.

The sequence types (ST) for the bacteria were identified using the *E. coli* Multi-locus Sequence Typing (MLST) scheme, MLST v.2.19.0 (https://github.com/tseemann/mlst; accessed 27 December 2023), at 100% identity. Genomes were screened against the seven housekeeping genes (*adk*, *fumC*, *gyrB*, *icd*, *mdh*, *purA*, *recA*) and assigned an ST that matched specific allele profiles. Based on a concatenated alignment of these gene sequences obtained from WGS using the neighbour-joining (NJ) method, a phylogenetic tree with 1000 bootstrap replicates was constructed. The evolutionary distances were calculated using the maximum composite likelihood method. All positions containing gaps and missing data were excluded from each sequence pair.

The assembled genomes were annotated using Prokka v.1.14.6 [65] and used as input for pangenome analysis (the totality of genes of all strains in the dataset) using Panaroo v1.2.7 (https://github.com/gtonkinhill/panaroo; accessed on 2 January 2024). A core genome sequence alignment of 3060 core genes (i.e., gene families present in 99% of genomes) was generated. Single-nucleotide polymorphisms (SNPs) were extracted from the core genome alignment using SNP-sites v.2.5.1 [65]. The core SNP alignment was used as input for building a maximum likelihood phylogenetic tree using RAxML v.8.2.12 [65] with a generalised time reversible (GTR [66] model of nucleotide substitution and gamma distribution of rate heterogeneity. The phylogeny was annotated and visualised using iTOL [67].

The serotyping of the *E. coli* isolates was performed using SerotypeFinder online tool (https://cge.food.dtu.dk/services/SerotypeFinder/, accessed on 2 January 2024). The genome assemblies were screened for the presence of AMR genetic determinants using the AMRFinderPlus v.3.10.23 software (https://github.com/ncbi/amr#ncbi-antimicrobial-resistance-gene-finder-amrfinderplus; accessed on 27 January 2024) and the ResFinder online tool (https://cge.food.dtu.dk/services/ResFinder; accessed on 15 March 2024), with identity thresholds of 85% and 98%, respectively. Additionally, a plasmid incompatibility database from PlasmidFinder (https://cge.food.dtu.dk/services/PlasmidFinder/, accessed on 17 March 2024) was utilised to identify and profile plasmids. Using the Complex Heatmap package in R, AMR phenotypes, genes, and plasmid replicons were compared across the human and environmental compartments.

### 4.5. Data Analysis

The data were organised in Excel^®^ 21 spreadsheets and then transferred to STATA^®^ 22 for analysis. We used R version 3.3.2 to examine the rates of resistance (R%), intermediate (I%), and susceptibility (S%), as well as the link between antibiotics and the phylogroups.

## 5. Conclusions

This study highlights the significance of genetic diversity and AMR of *E. coli* isolates from Lusaka’s clinical and environmental sources. The high prevalence of ESBL-producing *E. coli*, particularly strains harbouring *bla*_CTX-M-15_, emphasises the urgent need for effective surveillance and interventions to address the growing threat of antibiotic resistance in healthcare and environmental settings. Implementing robust infection control measures and antimicrobial stewardship programmes in healthcare settings is vital to mitigate the spread of these resistant pathogens and protect the public. This study had some limitations. First, the sample size was relatively small and restricted to one geographical region, which may limit the generalisability of the findings. Future research should expand the sample size and geographical scope to enhance understanding of AMR in clinical and environmental contexts. Mobile genetic element analysis should be incorporated to assess their role in transmitting resistance genes. Furthermore, developing and implementing robust antimicrobial stewardship programmes and policies is critical for mitigating the spread of antibiotic resistance in Zambia and beyond.

## Figures and Tables

**Figure 1 antibiotics-13-00951-f001:**
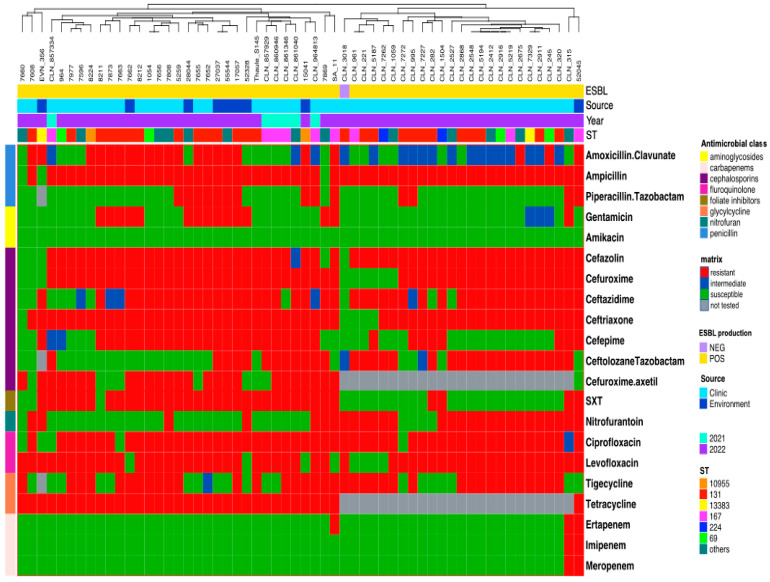
Matrix for antimicrobial susceptibility of *E. coli* genomes. The top row annotation shows the source, ST, and ESBL producers. The left row annotation refers to the antibiotic classes of antimicrobials. Columns are clustered using the Euclidean method.

**Figure 2 antibiotics-13-00951-f002:**
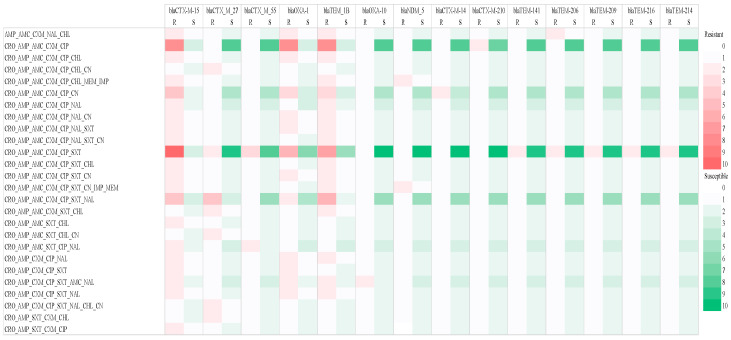
Heat map indicating the abundance of phenotypically resistant AMR genes and β-lactamase production in ESBL-producing *E. coli* strains from clinical and environmental isolates. Darker colours indicate stronger correlations, while lighter colours represent weaker correlations. Key: AMP (ampicillin); AMC (amoxicillin-clavulanic acid); CXM (cefuroxime); NAL (nalidixic acid); CHL (chloramphenicol); CRO (ceftriaxone); CIP (ciprofloxacin); SXT (trimethoprim-sulfamethoxazole); CN (gentamicin); MEM (meropenem); IMP (imipenem).

**Figure 3 antibiotics-13-00951-f003:**
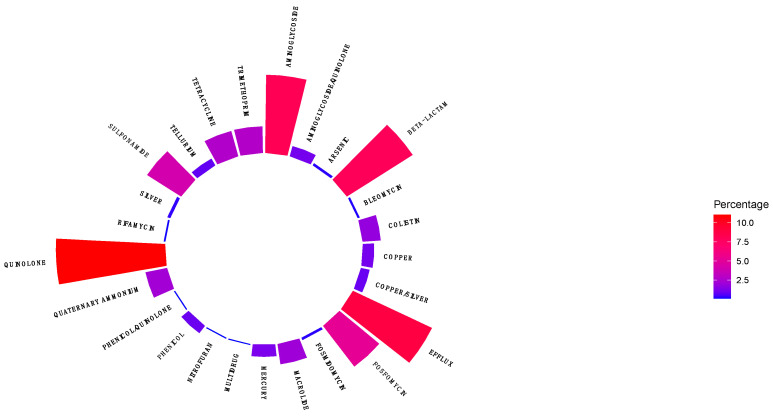
Distribution of AMR genes in antibiotic classes in ESBL-producing *E. coli* obtained from clinical and environmental sources.

**Figure 4 antibiotics-13-00951-f004:**
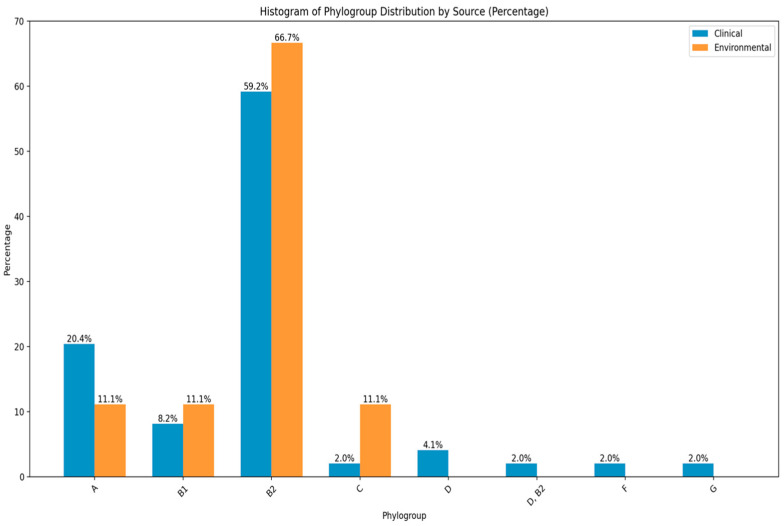
The distribution of phylogroups amongst environmental and clinical isolates of *E. coli*.

**Figure 5 antibiotics-13-00951-f005:**
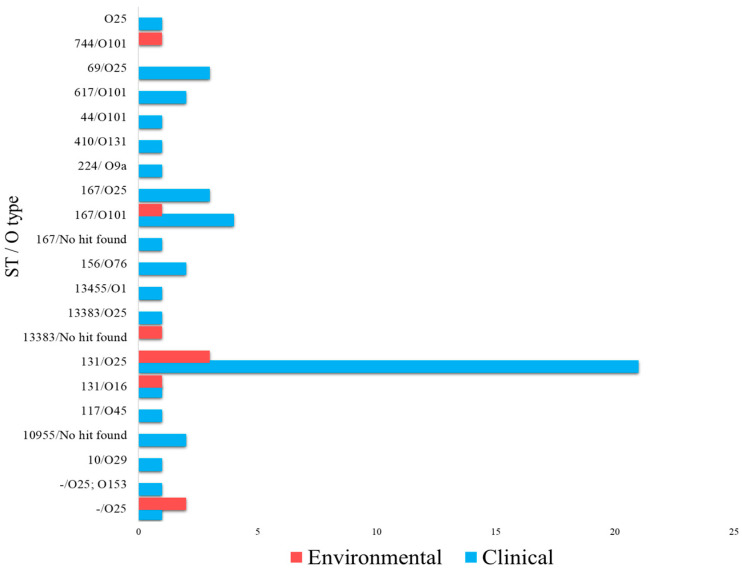
The diversity of phylogroups and sequence types in ESBL *E. coli* isolates.

**Table 1 antibiotics-13-00951-t001:** Characterisation of AMR genes and phylogroup in ESBL-producing *E. coli* strains.

Resistant Phenotype	Phylogroup	
	Environment	Clinical
AMP_AMC_CXM_NAL_CHL		B2
CRO_AMP_AMC_CXM_CIP	B2	A, B2, F
CRO_AMP_AMC_CXM_CIP_CHL		B2
CRO_AMP_AMC_CXM_CIP_CHL_CN		A
CRO_AMP_AMC_CXM_CIP_CHL_MEM_IMP	A	
CRO_AMP_AMC_CXM_CIP_CN	B2	A, B2
CRO_AMP_AMC_CXM_CIP_NAL		A, B1
CRO_AMP_AMC_CXM_CIP_NAL_SXT	C	
CRO_AMP_AMC_CXM_CIP_NAL_SXT_CN		A
CRO_AMP_AMC_CXM_CIP_SXT		A, B2, C, D, G
CRO_AMP_AMC_CXM_CIP_SXT_CHL		B2
CRO_AMP_AMC_CXM_CIP_SXT_CN		B2
CRO_AMP_AMC_CXM_CIP_SXT_CN_IMP_MEM		C
CRO_AMP_AMC_CXM_CIP_SXT_NAL	B2	A, B2
CRO_AMP_AMC_CXM_SXT_CHL		B1
CRO_AMP_AMC_SXT_CHL	B1	
CRO_AMP_AMC_SXT_CHL_CN		B1
CRO_AMP_AMC_SXT_CIP_NAL		A, C
CRO_AMP_CXM_CIP_NAL		B2
CRO_AMP_CXM_CIP_SXT		B2
CRO_AMP_CXM_CIP_SXT_AMC_NAL		B1, B2
CRO_AMP_CXM_CIP_SXT_NAL	B2	
CRO_AMP_CXM_CIP_SXT_NAL_CHL_CN		B2
CRO_AMP_SXT_CXM_CHL		B2
CRO_AMP_SXT_CXM_CIP	B2	

Key: AMP (ampicillin); AMC (amoxicillin-clavulanic acid); CXM (cefuroxime); NAL (nalidixic acid); CHL (chloramphenicol); CRO (ceftriaxone); CIP (ciprofloxacin); SXT (trimethoprim-sulfamethoxazole); CN (gentamicin); MEM (meropenem); IMP (imipenem).

**Table 2 antibiotics-13-00951-t002:** Antibiotic susceptibility patterns in *E. coli* genes.

Antibiotic	Clinical Sources			Environmental Sources			
	Resistant	Intermediate	Susceptible	Resistant	Intermediate	Susceptible	*p*-Value
Tetracyclines	4 (3.39%)	2 (1.69%)	28 (23.72%)	2 (1.69%)	0 (0%)	47 (39.83%)	0.07
Chloramphenicol	0 (0%)	1 (0.84%)	33 (27.96%)	1 (0.84%)	5 (4.24%)	43 (36.44%)	0.33
Aminoglycosides	15 (12.71%)	8 (6.78%)	11 (9.32%)	14 (11.86%)	11 (9.32%)	22 (18.64%)	0.71
Trimethoprim	25 (21.18%)	0 (0%)	9 (7.62%)	20 (16.94%)	2 (1.69%)	20 (16.94%)	0.24
Fluoroquinolones	0 (0%)	1 (0.84%)	33 (27.96%)	0 (0%)	0 (0%)	48 (40.67%)	0.32
Lincosamide Streptogramins’ B	3 (2.54%)	12 (10.17%)	19 (16.1%)	2 (1.69%)	13 (11.01%)	33 (27.69%)	0.33
Sulphonamide	18 (15.25%)	2 (1.69%)	14 (11.86%)	17 (14.4%)	12 (10.17%)	10 (8.47%)	0.07
Disinfectants	1 (0.84%)	1 (0.84%)	32 (27.12%)	1 (0.84%)	0 (0%)	48 (40.67%)	0.41

**Table 3 antibiotics-13-00951-t003:** Distribution of detected AMR genes in clinical and environmental *E. coli* isolates.

AMR Genes	Overall n (%)	Environmental n (%)	Clinical n (%)
Trimethoprim			
*dfrA*12	8 (13.8)	1 (12.5)	7 (87.5)
*dfrA*14	10 (17.2)	0 (0.0)	10 (100.0)
*dfrA*1	1 (1.7)	0 (0.0)	1 (100.0)
*dfrA*17	29 (50.0)	6 (20.7)	23 (79.3)
*dfrA*27	1 (1.7)	1 (100.0)	0 (0.0)
*dfrB*4	1 (1.7)	0 (0.0)	1 (100.0)
Sulphonamides			
*sul1*	37 (63.7)	7 (18.9)	30 (81.1)
*sul2*	42 (72.4)	7 (16.7)	35 (83.3)
*sul3*	0 (0.0)	0 (0.0)	0 (0.0)
Tetracycline			
*tet (A)*	37 (63.7)	6 (16.2)	31 (83.2)
*tet (B)*	17 (29.3)	1 (5.9)	16 (94.1)
*tet (M)*	0 (0.0)	0 (0.0)	0 (0.0)
Acquired quinolone resistance			
*qnrB1*	3 (5.1)	0 (0.0)	3 (100)
*qnrS1*	3 (5.1)	2 (66.7)	1 (33.3)
*qnrB6*	0 (0.0)	0 (0.0)	0 (0.0)
*aac (6′)-ib-cr5*	19 (32.7)	3 (15.8)	16 (84.2)
MLSB			
*mdf(A)*	58 (100.0)	9 (17.6)	42 (82.4)
*erm(B)*	5 (8.6)	0 (0.0)	5 (100.0)
*mph(A)*	38 (65.8)	6 (16.2)	31 (83.8)
Phenicols			
*catA1*	8 (15.4)	1 (12.5)	7 (87.5)
*catB3*	8 (15.4)	0 (0.0)	8 (100.0)
*floR*	1 (1.9)	0 (0.0)	1 (100.0)
*cmlA1*	1 (1.9)	0 (0.0)	1 (100.0)
Disinfectants			
*qacE*	43 (82.7)	8 (19.0)	34 (81.0)
*sitABCD*	40 (76.9)	7 (17.9)	32 (82.1)

**Table 4 antibiotics-13-00951-t004:** Genotypic characterisation of β-lactam AMR genes in ESBL-producing *E. coli* strains.

Beta-Lactam	Urine	Pus	Blood	Water	Meat	Equipment	Vegetables
*bla* _CTM-X-14_	1 (1.7%)	0 (0.0%)	0 (0.0%)	0 (0.0%)	0 (0.0%)	0 (0.0%)	0 (0.0%)
*bla* _CTM-X-15_	20 (34.5%)	6 (10.4%)	1 (1.7%)	2 (3.5%)	1 (1.7%)	1 (1.7%)	3 (5.2%)
*bla* _CTM-X-55_	1 (1.7%)	1 (1.7%)	0 (0.0%)	0 (0.0%)	0 (0.0%)	0 (0.0%)	0 (0.0%)
*bla* _CTM-X-27_	8 (13.8%)	1 (1.7%)	0 (0.0%)	0 (0.0%)	0 (0.0%)	1 (1.7%)	0 (0.0%)
*bla* _OXA-1_	15 (25.9%)	1 (1.7%)	0 (0.0%)	1 (1.7%)	0 (0.0%)	1 (1.7%)	1 (1.7%)
*bla* _TEM-1_	16 (27.6%)	2 (3.5%)	0 (0.0%)	2 (3.5%)	0 (0.0%)	2 (3.5%)	1 (1.7%)
*bla* _TEM_	2 (3.5%)	0 (0.0%)	0 (0.0%)	0 (0.0%)	0 (0.0%)	0 (0.0%)	0 (0.0%)
*bla* _TEM-1_	16 (27.6%)	2 (3.5%)	0 (0.0%)	2 (3.5%)	0 (0.0%)	2 (3.5%)	1 (1.7%)
*bla* _EC_	35 (60.4%)	12 (20.7%)	2 (3.5%)	2 (3.5%)	1 (1.7%)	3 (5.2%)	3 (5.2%)
*bla* _NDM-5_	0 (0.0%)	1 (1.7%)	0 (0.0%)	1 (1.7%)	0 (0.0%)	0 (0.0%)	0 (0.0%)

## Data Availability

The data supporting the reported results can be made available on request from the corresponding author.

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
