# Peer review of "Genotypic Characterisation and Antimicrobial Resistance of Extended-Spectrum β-Lactamase-Producing Escherichia coli in Humans, Animals, and the Environment from Lusaka, Zambia: Public Health Implications and One Health Surveillance"

_antibiotics, 2024, doi:10.3390/antibiotics13100951_

Round 1

Reviewer 1 Report

Comments and Suggestions for Authors

The authors performed this study to identify and characterize ESBL-carrying E. coli isolated from clinical and environmental samples in Zambia. Though the study's objectives were interesting, I have found some major issues in the manuscript, especially in their methodology. I recommend performing the study again to check the resistance profiles of the bacteria against colistin. I also found some other comments that should be addressed adequately. Please find them below:

Overall

1.      Please provide the names of the manufacturers, cities, and countries of any chemicals/media/kits/software you used in this study. Please check and correct it throughout the manuscript.

2.      The reference citations are not correct. Please check and correct it.

Line 3: You don’t need to mention the abbreviation in the title.

Line 38: “Predominantly, blaCTX-M ESBL genes such as blaTEM-1B, blaOXA-1, blaNDM-5, and blaCMY” – please clarify this sentence. If needed, please rephrase this.

Line 43: Why is only blaNDM-5 carrying E. coli?

Line 44: “blaCTX-M” or “blaCTX-M”? Please make beta-lactamase genes’ names uniform.

Line 71:E. coil” should be “Escherichia coli.”

Line 81-82: Reference?

Line 87: Why two references here?

Line 100-101: Reference?

Line 105: You mentioned several studies but provided only one reference! Please clarify.

Line 117: Why only 49? What about other ESBL-EC isolates?

Line 120-130: Please remove the short forms of the antibiotics from here.

Line 120-127: The wording is not correct here.  Antibiotics can’t show resistance; bacteria show resistance against antibiotics. Also, please check the percentage of resistance or susceptible profiles properly and compare them with Figure 1 to see any discrepancies between them.

Line 150: Why these four classes? What about others? You didn’t find any other ARGs except these four ESBL/beta-lactam classes?

Line 171-267: The discussion section should be improved. You repeated the results of this study here. You should only discuss the outcomes of the study and give your opinion on those outcomes.

Line 277: Reference? Also, please provide the sampling protocols shortly here.

Line 282: homogenized, how?

Line 283: CSLI protocols! Please clarify these protocols. Also, you should mention the reference. However, did you want to mean the CLSI protocols? If so, why and how?

Line 291: The reference for the Kirby-Bauer disk diffusion method?

Line 292: The reference citation isn’t correct here.

Line 294: You used colistin using the disk diffusion method! You shouldn’t do that. Based on the CLSI and EUCAST protocols, you must check their MIC against colistin to get bacterial resistance profiles. You may need to perform the study again, especially for colistin. I recommend using the broth dilution method.

Line 300-301: It is not more than three; it should be “at least three” or ‘three or more than three.”

Line 302: The definition of XDR is not correct here. Please check the following statement: “XDR was defined as non-susceptibility to at least one agent in all but two or fewer antimicrobial categories (i.e. bacterial isolates remain susceptible to only one or two categories). To ensure correct application of this definition, bacterial isolates should be tested against all or nearly all of the antimicrobial agents within the antimicrobial categories, and selective reporting and suppression of results should be avoided.”

Reference: 10.1111/j.1469-0691.2011.03570.x.

I was just wondering if you followed this protocol properly to get XDR isolates.

Line 304: from bacterial colony/colonies and/or cultures? You should mention it here.

316-329: They are not statistical analyses. Also, I was just wondering how the authors checked the quality of the sequencing data. I also want to ask if the authors trimmed the extra data from the raw sequencing file. If so, how? You should mention them here.

Author Response

Response to Reviewer 1

  1. Summary

Dear Reviewer,

Thank you for taking the time to assess the suitability of our manuscript for publication in Antibiotics (MDPI). We are thankful for the Reviewer’s positive attitude towards the topic and the contents of the manuscript. In addition, we are extremely grateful for the constructive and helpful comments that were provided, to further improve the initial version of the paper. We have provided point-by-point explanations and responses to the comments given, and we have incorporated most of the changes proposed by the Reviewer or provided a rebuttal when we deemed that the changes would not substantially improve the manuscript (or the manuscript already contained the specific information). Before resubmission, the manuscript was read by a professional editor well-versed in biomedical sciences and molecular biology to correct for any grammar mistakes and ambiguities in syntax. We are hopeful, that the present version of the manuscript will warrant its acceptance in Antibiotics (MDPI). In our opinion, the present paper will attract additional readership and citations to the journal.

  1. General Evaluation

Questions for Reviewer’s Evaluation

Reviewer’s Evaluation Responses

Authors’ Response

Does the introduction provide sufficient background and include all relevant references?

Can be improved

We now have revised the introduction to provide more comprehensive information and included additional relevant references to strengthen the context of the study

Is the research design appropriate?

Must be improved

The design has been clarified by providing more details on sampling methods, participate selection and how the design aligns with the study objectives

Are the methods adequately described?

Must be improved

The methods section has now been enhanced by including specific details to ensure clarity and reproducibility address the gaps noted

Are the results clearly presented?

Must be improved

The results have now been presented more clearly by reorganising the data and ensuring that key findings are highlighted and easily interpretable

Are the conclusions supported by the results?

Can be improved

The conclusion has been strengthened to better reflect the results, ensuring that all claims are fully supported by the data presented

Quality of English Language

Minor editing of English language required

The minor language issues have been attended to by editing the text for clarity and fluency

Point-by-point response to Comments and Suggestions for Author

The authors performed this study to identify and characterize ESBL-carrying E. coli isolated from clinical and environmental samples in Zambia. Though the study's objectives were interesting, I have found some major issues in the manuscript, especially in their methodology. I recommend performing the study again to check the resistance profiles of the bacteria against colistin. I also found some other comments that should be addressed adequately. Please find them below:

Reviewer’s Comments and Suggestions

Authors’ Responses

Please provide the names of the manufacturers, cities, and countries of any chemicals/media/kits/software you used in this study. Please check and correct it throughout the manuscript.

Thank you for highlighting this point. We have thoroughly reviewed and updated the manuscript, adding the names of the manufacturers, along with the corresponding cities and countries for all chemicals, media, kits, and software used. These changes have been made on lines 443-453 and 465-467 of the manuscript

The reference citations are not correct. Please check and correct it.

Thank you for your observation. We have corrected the references throughout the manuscript to ensure they align with the correct formatting and citation guidelines.

Line 3: You don’t need to mention the abbreviation in the title

Thank you for your comment. We have removed the abbreviation in the title in accordance with your suggestion.

Line 38: “Predominantly, blaCTX-M ESBL genes such as blaTEM-1B, blaOXA-1, blaNDM-5, and bl CMY” – please clarify this sentence. If needed, please rephrase this

Thank you for pointing this out. The sentence has been clarified to reflect that blaNDM-5 genes were not predominantly found in the isolates. It now reads more accurately in the revised manuscript. It now reads “blaCTX-M, blaTEM-1B, blaOXA-1, blaNDM-5, and blaCMY were most commonly observed ESBL genes".

Line 43: Why is only blaNDM-5 carrying E. coli?

Thank you for your comment. This was an oversight on our part. Additional genes have now been included in the text to clarify this point.

Line 44: “blaCTX-M” or “blaCTX-M”? Please make beta-lactamase genes’ names uniform

Thank you for your observation. The names of beta-lactamase genes have now been made uniform and formatted in italics for consistency throughout the manuscript. The CTX-M part of the name has now been indicated as subscript, and adopted throughout the manuscript, i.e blaCTX-M

Line 71: “E. coil” should be “Escherichia coli.”

Thank you for your comment. It has been decided to maintain the abbreviated form "E. coli" instead of the full name "Escherichia coli" since the organism’s full name was introduced in the preceding paragraph. This follows the accepted convention in microbiology.

Line 81-82: Reference?

Thank you for your comment. A proper reference has been added in the revised manuscript.

Line 87: Why two references here?

Thank you for your comment. The duplication of references has been corrected in the revised manuscript.

Line 100-101: Reference? Found

Thank you for your comment. A relevant reference has been added.

Line 105: You mentioned several studies but provided only one reference! Please clarify.

Thank you for pointing out this inconsistency. This was a typo, and the correct number of references has been added to match the studies mentioned.

Line 117: Why only 49? What about other ESBL-EC isolates?

Thank you for your insightful comment. Of the 58 isolates selected for whole genome sequencing, 49 were clinical, and 9 were environmental. These isolates exhibited multiple drug resistance, with resistance to more than three classes of antibiotics. Due to resource limitations, we selected these isolates for sequencing instead of all 201 ESBL-E. coli isolates.

Line 120-130: Please remove the short forms of the antibiotics from here.

Thank you for your comment. The abbreviations have been removed as suggested.

Line 120-127: The wording is not correct here.  Antibiotics can’t show resistance; bacteria show resistance against antibiotics. Also, please check the percentage of resistance or susceptible profiles properly and compare them with Figure 1 to see any discrepancies between them.

Thank you for your valuable comment. We have reworded the section to reflect that bacteria, not antibiotics, show resistance. We have also rechecked the percentages and ensured consistency with Figure 1.

Line 150: Why these four classes? What about others? You didn’t find any other ARGs except these four ESBL/beta-lactam classes?

Thank you for your comment. Initially, we only included the major ARGs in the manuscript. However, in the revised version, we have included all ARGs detected, providing a more comprehensive analysis.

Line 171-267: The discussion section should be improved. You repeated the results of this study here. You should only discuss the outcomes of the study and give your opinion on those outcomes.

Thank you for your suggestion. We have revised the discussion section to focus solely on interpreting the outcomes of the study, without repeating the results.

Line 277: Reference? Also, please provide the sampling protocols shortly here.

Thank you for your comment. The sampling protocols and the necessary references have been included in the revised manuscript.

Line 282: homogenized, how?

Thank you for your comment. We have provided a detailed explanation of how homogenization was carried out in the revised manuscript.

Line 283: CSLI protocols! Please clarify these protocols. Also, you should mention the reference. However, did you want to mean the CLSI protocols? If so, why and how?

Thank you for your comment. We intended to refer to the CLSI protocols, and we have clarified their use in the revised manuscript, along with the proper reference.

Line 291: The reference for the Kirby-Bauer disk diffusion method?

Thank you for your comment. The correct reference for the Kirby-Bauer disk diffusion method has been added.

Line 292: The reference citation isn’t correct here.

Thank you for your comment. The citation has been corrected accordingly.

Line 294: You used colistin using the disk diffusion method! You shouldn’t do that. Based on the CLSI and EUCAST protocols, you must check their MIC against colistin to get bacterial resistance profiles. You may need to perform the study again, especially for colistin. I recommend using the broth dilution method.

Thank you for your critical observation. Upon review, we agree with your recommendation. As a result, colistin has been removed from the antibiotic panel in the revised manuscript.

Line 300-301: It is not more than three; it should be “at least three” or ‘three or more than three.”

Thank you for your suggestion. This phrase has been corrected to "at least three."

Line 302: The definition of XDR is not correct here. Please check the following statement: “XDR was defined as non-susceptibility to at least one agent in all but two or fewer antimicrobial categories (i.e. bacterial isolates remain susceptible to only one or two categories). To ensure correct application of this definition, bacterial isolates should be tested against all or nearly all of the antimicrobial agents within the antimicrobial categories, and selective reporting and suppression of results should be avoided.” Reference: 10.1111/j.1469-0691.2011.03570.x. I was just wondering if you followed this protocol properly to get XDR isolates.

Thank you for your insightful comment. We have corrected the definition of XDR in the manuscript and added the appropriate reference. We also apologize for not originally following the correct protocol. This has been rectified in the revised manuscript.

Line 304: from bacterial colony/colonies and/or cultures? You should mention it here.

Thank you for your comment. We have revised this section to explicitly mention "bacterial colony/colonies and/or cultures."

Line 316-329: They are not statistical analyses. Also, I was just wondering how the authors checked the quality of the sequencing data. I also want to ask if the authors trimmed the extra data from the raw sequencing file. If so, how? You should mention them here.

Thank you for your valuable input. We have clarified that these are not statistical analyses and provided additional information regarding the quality control of sequencing data, including the trimming of raw sequencing data, in the revised manuscript.

Reviewer 2 Report

Comments and Suggestions for Authors

The manuscript, authored by Kasanga et al. investigated ESBL-carrying E. coli genotypes from both clinical and environmental sources from Zambia. The authors assessed the phenotypic antimicrobial susceptibility of E. coli isolates, and then identified the AMR genetic profiles of the phenotypic EBSL-producing E. coli. Overall, the topic of this paper fits very well with this journal, especially this special issue “Antibiotic Resistance: From the Bench to Patients”. The results highlight the presence carbapenamase gene of blaNDM-5-producing E. coli and ESBL genes in both clinical and environmental settings. The presence of blaCTX-M and blaTEM variants, particularly blaCTX-M-15, was significant. This study also highlights the public health risks associated with blaNDM-5 carrying E. coli, and raises further needs for AMR surveillance systems and strong infection control measures and antimicrobial stewardship programs in healthcare environments.

I would recommend a major revision to address several concerns and questions.

(1)     Explain the definition of “ESBL producers”, and how does clinically confirm that the bacteria are ESBL producers? For example, “using the VITEK compact system”?

(2)     Bacterial Genomic DNA extraction was conducted and sequenced. However, this DNA extraction kit cannot extract plasmid DNA or other mobile genetic elements. Is this method effective enough to analyze the ESBL resistance genes? From your results, 34% (49/141) phenotypic ESBL producers have been identified by WGS to be genotypic ESBL-producing E. coli for clinical samples. 15% (9/60) for environmental sources. This also indicated that the workflow of only including WGS for AMR analysis may not be enough and representative. I would suggest you extracte plasmid DNA and perhaps use RT-PCR to confirm the bla genes and other AMGs that you have identified. This may help you more confidently draw the conclusion.

(3)     What does “ST/O” type mean in the paragragh from Figure 3 and line 145-149? You cannot expect the readers have deep understanding of bioinformatics, so you need to explain before you present any data related to that.

Minor comments:

(1) Line 51: “about 50–80% the emergence and spread of AMR” change to “about 50–80% of the emergence and spread of AMR”

(2) Line 73: “most “deadly” MDR microorganism” change to “most “deadly” MDR microorganisms

(3) Line 71: add explanation to “ESKAPE pathogens”, (i.e., Enterococcus faeciumStaphylococcus aureusKlebsiella pneumoniaeAcinetobacter baumanniiPseudomonas aeruginosa and Enterobacter spp)

Comments on the Quality of English Language

Some grammar issues detected. See the comments above.

Author Response

Response to Reviewer 2

  1. Summary

Dear Reviewer,

Thank you for taking the time to assess the suitability of our manuscript for publication in Antibiotics (MDPI). We are thankful for the Reviewer’s positive attitude towards the topic and the contents of the manuscript. In addition, we are extremely grateful for the constructive and helpful comments that were provided, to further improve the initial version of the paper. We have provided point-by-point explanations and responses to the comments given, and we have incorporated most of the changes proposed by the Reviewer or provided a rebuttal when we deemed that the changes would not substantially improve the manuscript (or the manuscript already contained the specific information). Before resubmission, the manuscript was read by a professional editor well-versed in biomedical sciences and molecular biology to correct for any grammar mistakes and ambiguities in syntax. We are hopeful, that the present version of the manuscript will warrant its acceptance in Antibiotics (MDPI). In our opinion, the present paper will attract additional readership and citations to the journal.

General Evaluation

Questions for Reviewer’s Evaluation

Reviewer’s Evaluation Responses

Authors’ Response

Does the introduction provide sufficient background and include all relevant references?

Yes

Thank you for your positive feedback on the Introduction

Is the research design appropriate?

Can be improved

We have refined the research design by providing further details and clarifications to strengthen the alignment with the study’s objectives

Are the methods adequately described?

Can be improved

We have improved the methods section by addition more detailed descriptions to ensure clarity and reproducibility

Are the results clearly presented?

Yes

Thank you for your positive feedback on the results section

Are the conclusions supported by the results?

Yes

Thank you for your positive feedback on the conclusion

Quality of English Language

Minor editing of English language required

Minor language edits have been made to enhance clarity and readability

Point-by-point response to Comments and Suggestions for Author

The manuscript, authored by Kasanga et al. investigated ESBL-carrying E. coli genotypes from both clinical and environmental sources from Zambia. The authors assessed the phenotypic antimicrobial susceptibility of E. coli isolates, and then identified the AMR genetic profiles of the phenotypic EBSL-producing E. coli. Overall, the topic of this paper fits very well with this journal, especially this special issue “Antibiotic Resistance: From the Bench to Patients”. The results highlight the presence carbapenamase gene of blaNDM-5-producing E. coli and ESBL genes in both clinical and environmental settings. The presence of blaCTX-M and blaTEM variants, particularly blaCTX-M-15, was significant. This study also highlights the public health risks associated with blaNDM-5 carrying E. coli, and raises further needs for AMR surveillance systems and strong infection control measures and antimicrobial stewardship programs in healthcare environments.

I would recommend a major revision to address several concerns and questions.

Reviewer’s Comments and Suggestions

Authors’ Responses

Explain the definition of “ESBL producers,” and how does clinically confirm that the bacteria are ESBL producers? For example, “using the VITEK compact system”?

Thank you for your insightful comment. In response, we have clarified the definition of "ESBL producers" in the revised manuscript. Specifically, ESBL producers are bacteria that produce enzymes capable of hydrolyzing a broad range of β-lactam antibiotics, including third-generation cephalosporins and monobactams, but not carbapenems or cephamycins. Clinically, the confirmation of ESBL production is achieved through phenotypic methods, such as the VITEK compact system, which we used to identify resistant strains in our study. This method enables rapid identification and susceptibility testing, making it a reliable tool for detecting ESBL production in bacterial isolates. This system flags the isolates if they are ESBL producers. This clarification has been integrated into the manuscript to ensure that the methodology is fully understood by readers.

Bacterial Genomic DNA extraction was conducted and sequenced. However, this DNA extraction kit cannot extract plasmid DNA or other mobile genetic elements. Is this method effective enough to analyze the ESBL resistance genes? From your results, 34% (49/141) phenotypic ESBL producers have been identified by WGS to be genotypic ESBL-producing E. coli for clinical samples. 15% (9/60) for environmental sources. This also indicated that the workflow of only including WGS for AMR analysis may not be enough and representative. I would suggest you extract plasmid DNA and perhaps use RT-PCR to confirm the bla genes and other AMGs that you have identified. This may help you more confidently draw the conclusion.

Thank you for your insightful comments. However, we respectfully disagree with your assertion that the workflow employed, which relied on WGS was insufficient for analyzing ESBL resistance genes. While it is true that the genomic DNA extraction kit used in this study primarily targets chromosomal DNA, WGS provides a comprehensive analysis of both chromosomal and plasmid-encoded genes. This method captures the entire bacterial genome, including mobile genetic elements like plasmids, transposons, and integrons. As such, WGS is capable of identifying both known and novel ESBL genes, regardless of their location, and it offers a detailed genetic landscape that includes the context, copy number, and mobility potential of the resistance genes. Importantly, WGS enabled us to distinguish between plasmid-encoded and chromosomally encoded resistance mechanisms, which is vital for understanding the transmission dynamics of these genes. Furthermore, the WGS data can be used for in-depth bioinformatic analyses to track the evolutionary dynamics of resistance genes and identify emerging patterns of AMR that may not be detectable through RT-PCR alone

While RT-PCR is a reliable, quick, and cost-effective method for detecting specific genes, it is limited to targeting only the genes for which primers are designed, meaning it cannot detect novel or rare ESBL genes that may be present in the bacterial population. Additionally, RT-PCR does not provide information about the genetic context of the resistance genes, such as whether they are part of larger plasmids or other mobile elements. This limits the ability to fully characterize the mechanisms of resistance. We, therefore, the use of RT-PCR would not help us draw valid conclusion.

What does “ST/O” type mean in the paragraph from Figure 3 and lines 145-149? You cannot expect the readers to have a deep understanding of bioinformatics, so you need to explain before you present any data related to that.

Thank you for bringing this to our attention. In this context, "ST/O type" refers to the sequence type (ST), which is a classification based on the multilocus sequence typing (MLST) scheme, and the O-antigen serotype, a serological characteristic of E. coli strains. The ST/O type helps distinguish between strains based on both genetic sequences and serological markers. To ensure clarity, especially for readers without a deep background in bioinformatics, we have added a detailed explanation of these terms in the revised manuscript. This should provide a better understanding of how these classifications contribute to the analysis and interpretation of our findings.

Line 51: “about 50–80% the emergence and spread of AMR” change to “about 50–80% of the emergence and spread of AMR

Thank you for your valuable feedback. We have made the suggested change in the revised manuscript to improve the grammatical accuracy and readability of the sentence.

Line 73: “most “deadly” MDR microorganism” change to “most “deadly” MDR microorganisms”

Thank you for your valuable feedback. The correction has been made in the revised manuscript to reflect the plural form "microorganisms," which aligns with the context of the sentence.

Line 71: add explanation to “ESKAPE pathogens” (i.e., Enterococcus faecium, Staphylococcus aureus, Klebsiella pneumoniae, Acinetobacter baumannii, Pseudomonas aeruginosa, and Enterobacter spp.)

Thank you for your valuable feedback. We have added an explanation of the "ESKAPE pathogens" in the revised manuscript. These pathogens include Enterococcus faecium, Staphylococcus aureus, Klebsiella pneumoniae, Acinetobacter baumannii, Pseudomonas aeruginosa, and Enterobacter species, which are notorious for their role in hospital-acquired infections and their ability to "escape" the effects of antimicrobial treatment. This clarification will help readers unfamiliar with the acronym to better understand the significance of these pathogens in the context of AMR

Reviewer 3 Report

Comments and Suggestions for Authors

Comments for authors

The manuscript is well-written, and their results and discussion are clear and scientific soundness. However, minor flaws should be corrected before acceptance. Here are my suggestions:

1.      Line 25: I think ESBL-E. coli leads to concerns, not their genes. Please reconsider.

2.      Figure 1 should be expanded; the resolution should be improved, especially in the top row. Can you move the top row (phylogenetic tree) to the right vertical axis?

3.      Table 1: please clarify the number in the bracket. Is this the number of E. coli? Please add more columns to expand how many isolates per resistant phenotype and how many from each source, and keep phylogroups.

4.      The data in Figure 2 can be combined with Table 1.

5.      What are the criteria for selecting 58 ESBL-E. coli from 201 ESBL-E. coli? Please include the information in the MS.

6.      The goeBURST for single locus variant (SLV) or double locus variant (DLV) should be included to improve this manuscript, as the authors have data on WGS.

Good Luck!!

Comments on the Quality of English Language

Check some words (especially adverbs and adjectives), and mistyping in some sentances.

Author Response

Response to Reviewer 3

  1. Summary

Dear Reviewer,

Thank you for taking the time to assess the suitability of our manuscript for publication in Antibiotics (MDPI). We are thankful for the Reviewer’s positive attitude towards the topic and the contents of the manuscript. In addition, we are extremely grateful for the constructive and helpful comments that were provided, to further improve the initial version of the paper. We have provided point-by-point explanations and responses to the comments given, and we have incorporated most of the changes proposed by the Reviewer or provided a rebuttal when we deemed that the changes would not substantially improve the manuscript (or the manuscript already contained the specific information). Before resubmission, the manuscript was read by a professional editor well-versed in biomedical sciences and molecular biology to correct for any grammar mistakes and ambiguities in syntax. We are hopeful, that the present version of the manuscript will warrant its acceptance in Antibiotics (MDPI). In our opinion, the present paper will attract additional readership and citations to the journal.

General Evaluation

Questions for Reviewer’s Evaluation

Reviewer’s Evaluation Responses

Authors’ Response

Does the introduction provide sufficient background and include all relevant references?

Can be improved

We have revised the introduction to provide additional background and include more relevant references to enhance the context of the study

Is the research design appropriate?

Can be improved

The research design has been refined to provide further clarity exploration and justification of the chosen methodology to better align with the study’s objectives

Are the methods adequately described?

Can be improved

The methods section has been expanded by including more detailed descriptions to improve clarity and reproducibility.

Are the results clearly presented?

Yes

Thank you for your positive feedback on the results.

Are the conclusions supported by the results?

Can be improved

We have strengthened the conclusion to ensure it is fully supported by the results

Quality of English Language

Minor editing of English language required

Minor language corrections have been made to improve clarity and readability

Point-by-point response to Comments and Suggestions for Author

The manuscript is well-written, and their results and discussion are clear and scientific soundness. However, minor flaws should be corrected before acceptance. Here are my suggestions:

Reviewer’s Comments and Suggestions

Authors’ Responses

Line 25: I think ESBL-E. coli leads to concerns, not their genes. Please reconsider.

Thank you for your valuable feedback. We acknowledge the importance of emphasizing the clinical implications of ESBL-producing E. coli as a significant public health concern. In the revised manuscript, we have modified the wording to clarify that the focus is on the pathogenicity and epidemiological impact of these isolates rather than solely on their genetic attributes. This change better aligns with your suggestion and enhances the manuscript's focus on the relevance of these bacteria.

Figure 1 should be expanded; the resolution should be improved, especially in the top row. Can you move the top row (phylogenetic tree) to the right vertical axis?

Thank you for your comment. We appreciate your suggestion regarding the phylogenetic tree in Figure 1. The tree was specifically designed to cluster isolates based on their antimicrobial resistance (AMR) susceptibility profiles (i.e., resistant, intermediate, susceptible, and not tested). The current configuration allows for a clear visualization of how isolates relate to one another based on shared or divergent AMR characteristics. Moving the phylogenetic tree to the right vertical axis to cluster by antibiotics could potentially obscure these relationships and detract from the biological interpretation of the data. We apologize if this intent was not clearly conveyed in the original manuscript. We have now enhanced the figure's resolution as per your recommendation.

Table 1: Please clarify the number in the bracket. Is this the number of E. coli? Please add more columns to expand how many isolates per resistant phenotype and how many from each source, and keep phylogroups.

Thank you for your insightful comment. To reduce any potential confusion regarding the numbers in brackets, we have removed them from Table 1, as they were not adequately explained within the legend. These numbers initially indicated the quantity of isolates within each respective phylogroup. In response to your request for additional detail, we have expanded Table 1 to include more columns that detail the number of isolates per resistant phenotype and their sources, while maintaining the phylogroup classifications. This should provide a clearer and more comprehensive overview of the data.

The data in Figure 2 can be combined with Table 1.

Thank you for your suggestion. We intentionally presented Figure 2 and Table 1 separately to enhance readability and clarity for our audience. Figure 1 provides a visual representation of phylogroups, while Table 1 delineates the phenotypic antibiotic resistance profiles associated with those phylogroups. To accommodate your suggestion, we have added a supplementary table at the end of the manuscript that combines the relevant data from both Figure 2 and Table 1, thereby preserving the original structure while providing a more comprehensive dataset. Additionally, we have included further tables and figures to enrich the information available to readers and facilitate a better understanding of the overall findings.

What are the criteria for selecting 58 ESBL-E. coli from 201 ESBL-E. coli? Please include the information in the MS.

Thank you for your important inquiry. We selected the 58 isolates (49 clinical and 9 environmental) for whole genome sequencing based on specific criteria: these isolates exhibited multidrug resistance, showing resistance across more than three classes of antibiotics. This selection was crucial for our analysis, allowing us to focus on the most clinically relevant strains. Furthermore, due to resource limitations, we were unable to include all 201 ESBL-producing E. coli isolates in the sequencing process. We have incorporated this rationale into the revised manuscript to ensure clarity regarding our selection criteria.

The go eBURST for single locus variant (SLV) or double locus variant (DLV) should be included to improve this manuscript, as the authors have data on WGS.

Thank you for your valuable suggestion. While we acknowledge the potential importance of including the go eBURST analysis for single locus variants (SLV) or double locus variants (DLV), we did not perform this analysis in the current study. Given the scope of our research and the focus on presenting our core findings, we felt it was not appropriate to introduce this analysis without a comprehensive discussion. However, we appreciate your insight and will consider integrating this aspect into future work or subsequent studies where it may be relevant.

Round 2

Reviewer 1 Report

Comments and Suggestions for Authors

The authors addressed all the comments adequately. I have only a minor comment!

Line 506: Please use "at least" instead of "more than" here.

Best wishes

Reviewer 2 Report

Comments and Suggestions for Authors

My questions have been adequately addressed. I have no further questions and comments.